# The Prevalence and Clinical Impact of Adenomyosis in Pregnancy-Related Hysterectomy

**DOI:** 10.3390/jcm11164814

**Published:** 2022-08-17

**Authors:** Michele Orsi, Edgardo Somigliana, Fulvia Milena Cribiù, Gianluca Lopez, Laura Buggio, Manuela Wally Ossola, Enrico Ferrazzi

**Affiliations:** 1Department of Woman, New-Born and Child, Fondazione Istituto di Ricovero e Cura a Carattere Scientifico Ca’ Granda Ospedale Maggiore Policlinico, Via della Commenda, 12, 20122 Milan, Italy; 2Department of Clinical Sciences and Community Health, Università degli Studi di Milano, Via Festa del Perdono, 7, 20122 Milan, Italy; 3Maternal-Infant Department, Azienda Socio Sanitaria Territoriale Rhodense, Garbagnate Hospital, Via Carlo Forlanini, 95, 20024 Milan, Italy; 4Pathology Unit, Fondazione Istituto di Ricovero e Cura a Carattere Scientifico Ca’ Granda Ospedale Maggiore Policlinico, Via Francesco Sforza, 35, 20122 Milan, Italy; 5Pathology Unit, Azienda Socio Sanitaria Territoriale Bergamo Ovest, Treviglio-Caravaggio Hospital, Piazzale Ospedale Luigi Meneguzzo, 1, 24047 Treviglio, Italy; 6Gynecology Unit, Fondazione Istituto di Ricovero e Cura a Carattere Scientifico Ca’ Granda Ospedale Maggiore Policlinico, Via della Commenda, 12, 20122 Milan, Italy

**Keywords:** adenomyosis, pregnancy-related hysterectomy, endometriosis, placenta previa, chorionamnionitis, preterm birth

## Abstract

Background: The epidemiology of adenomyosis has been traditionally based on patients undergoing hysterectomy for gynecological indications, while its prevalence among hysterectomies performed for obstetric complications is unknown. The aim of this study was to assess the prevalence and clinical impact of adenomyosis diagnosed through histology among women undergoing pregnancy-related hysterectomy (PH). Methods: This was a retrospective cohort study. Women who delivered at a tertiary care regional obstetric hub in Milan between 2009 and 2020 were reviewed to identify cases of PH. Histopathological reports of surgical specimens were examined. Cases with adenomyosis were compared to those without adenomyosis for baseline characteristics, obstetric history and outcomes. Results: During the study period there were 71,061 births and a total of 130 PH, giving a PH incidence of 1.83 per 1000 deliveries. Adenomyosis cases were 18, giving a prevalence of 13.8%. Adenomyosis was associated with placenta previa (77.8 vs. 45.5%, *p* = 0.01), chorionamnionitis (27.8 vs. 5.4%, *p* = 0.008), lower gestational age at birth (32 ± 4.6 vs. 35.5 ± 3.6 weeks’ gestation, *p* = 0.0004), and intrauterine fetal demise among twin pregnancies (50 vs. 4.5%, *p* = 0.048). Conclusion: Adenomyosis entails a relevant impact on obstetric and perinatal outcomes related to PH. More evidence is needed on the clinical relevance of an ultrasonographic diagnosis of adenomyosis before conception.

## 1. Introduction

Adenomyosis is a benign gynecological disorder defined by the ectopic presence of endometrial glands and stroma within the uterine myometrium that results hypertrophic and hyperplastic [1,2]. Related symptoms include heavy menstrual periods, dysmenorrhea, and chronic pelvic pain, resulting in a significant impact on women’s quality of life [3,4]. Furthermore, there is growing evidence that this condition entails a relevant impact on fertility and pregnancy outcomes through anatomical, functional and immunochemical alterations of the uterine environment [5,6,7,8,9,10,11].

Transvaginal ultrasound is the method of first choice for diagnosis and decision making, while magnetic resonance is needed only in selected cases [12,13,14]. Definitive confirmation is traditionally based on histological examination at hysterectomy [15,16]. During the last decade, ultrasound-guided and laparoscopic approaches for uterine biopsy have been investigated, reporting variable sensitivity and high specificity in detecting myometrial lesions including adenomyosis [17,18,19,20]. While this approach is promising, no conclusive recommendations have yet been published regarding the optimal sampling technique [17]. Therefore, most epidemiological data on adenomyosis derive from patients undergoing hysterectomy for gynecological indications [16,21]. 

To the best of our knowledge, no data are available regarding the prevalence of adenomyosis among hysterectomies performed for obstetric complications. 

Pregnancy-related hysterectomy (PH) is usually performed as a life-saving procedure for uncontrollable hemorrhage, placenta accreta spectrum disorders, uterine rupture, or sepsis [22]. The World Health Organization (WHO) identified PH as a maternal near-miss criterion for obstetric systems surveillance [23,24]. The analysis of these cases has been promoted to implement strategies aimed at improving peripartum safety [25].

In this manuscript, we present the results of a retrospective study conducted at a single high-clinical volume obstetric hub to investigate the frequency and clinical impact of adenomyosis in women who underwent hysterectomy for severe obstetric complications.

## 2. Materials and Methods

### 2.1. Study Population

We reviewed the medical records of all women who underwent PH between 2009 and 2020 at the Mangiagalli Centre, Fondazione IRCCS Ca’ Granda Ospedale Maggiore Policlinico. This is the largest tertiary-care regional obstetric hub in Lombardia, Northern Italy, and affiliated with the University of Milan.

PH was defined as the surgical removal of the uterus performed for obstetric complications within the first 42 days after delivery [22]. Histopathological reports of surgical specimens were reviewed. Cases with adenomyosis were compared to those without adenomyosis for baseline characteristics, obstetric and gynaecological history, index pregnancy information and outcomes.

The local Ethics Committee Milan Area 2 approved the research protocol (approval no. 35_2021, 14 January 2021). All participants’ rights were protected in agreement with the Good Clinical Practice and the Helsinki Declaration. 

### 2.2. Histopathological Analysis 

Standard processing of surgical specimens included a minimum of four full-thickness samples obtained from the anterior and posterior uterine walls of the uterine body and cervix. At least one of them systematically included the placental implant, in order to assess the endo-myometrial interface for suspected or unexpected abnormal placentation. Adenomyosis was diagnosed when the distance between the lower border of the endometrium and the affected myometrial area was over one-half of a low-power field (−2.5 mm) [26]. Additional myometrial evaluations were made if macroscopic inspection highlighted areas of pathologies. 

### 2.3. Statistical Analysis 

The Kolmogorov–Smirnov test was used to assess normality of continuous variables. Student’s *t*-test and the Mann–Whitney U test were chosen for the analysis of normally and non-normally distributed continuous variables, respectively. Chi-square analysis or the Fisher Exact test were used for categorical variables, as appropriate. All tests were two-sided, and *p* values lower than 0.05 were stated as statistically significant. IBM SPSS 22.0 (Armonk, NY, USA) software was used for statistical analyses.

## 3. Results 

During the 12-years study period, we recorded a total of 71,061 births and 133 associated hysterectomies. Three cases were excluded because the indication for elective postpartum hysterectomy was cervical cancer, leaving the study group consisting of 130 PH cases and giving an incidence of 1.83 per 1000 deliveries (Figure 1). 

Adenomyosis was diagnosed in 18 cases, resulting in a prevalence of 13.8%. Representative cases are illustrated in Figure 2 (macroscopic specimen) and Figure 3, Figure 4, Figure 5 and Figure 6 (histological findings). Table 1 shows baseline characteristics of women with and without adenomyosis. In 14 of the 18 affected cases the medical history was silent as regards a pre-pregnancy diagnosis of adenomyosis. Previous surgery for endometriosis was significantly more frequent in subjects with adenomyosis. Previous hysteroscopic metroplasty for uterine septum was also more frequent in the adenomyosis group. All cases were for incomplete septum. Regarding the indication, in two cases in the affected group, and in one in the unaffected group, it was multiple abortions. In the other two cases, one for each group, the indication was a large septum with a depth over 2 cm. One patient in the non-adenomyosis group had a history of hysteroscopic metroplasty and laparoscopic myomectomy, while no overlap between different types of previous pelvic surgery was observed in the adenomyosis group.

Table 2 illustrates pregnancy variables. Cases of PH with adenomyosis were significantly associated with assisted reproductive technologies and placenta previa. Among perinatal outcomes, adenomyosis was significantly associated with chorionamnionitis, lower gestational age at birth and intrauterine fetal demise in twin pregnancies.

## 4. Discussion

The prevalence of adenomyosis in pregnancy related hysterectomies in the study period of twelve years was 13.8%. In most cases, the medical history was silent as regards a pre-pregnancy diagnosis of the disease. Cases of PH with adenomyosis reported more frequently a history of endometriosis, assisted reproductive technologies, and surgery for uterine septum. Placenta previa, chorioamnionitis and earlier gestational age at delivery were also significantly associated with the presence of the disease.

To the best of our knowledge, this is the first study focusing on the clinical burden of adenomyosis among women undergoing hysterectomy for obstetric complications, providing the opportunity to investigate its impact from a new perspective. Moreover, as maternal near-miss criterion defined by WHO, the study of PH cases is expected to contribute to peripartum safety improvement [23,24].

Only four of the eighteen affected cases in this series reported previous history of adenomyosis or endometriosis, of whom one had surgery for adenomyosis. To explain the remaining 14 cases with silent history, two hypotheses should be mentioned. First, an antecedent diagnostic suspicion of adenomyosis at imaging might have been missed at the medical history interview at the time of delivery. Secondly, the presence of the disease was unknown before pregnancy. The diagnostic delay up to 10 years that almost invariably characterizes adenomyosis diagnosis, and the frequently inappropriate sonographic imaging of the myometrium support the latter option [21,27]. In addition, in pregnancy, the diagnosis may be missed because the sonographic imaging of the myometrium is altered by the neovascularization. If these hypotheses were true, the potential impact of our findings would be amplified. Failure to diagnose adenomyosis may prevent affected patients from being identified as high-risk obstetric cases, as such denying the access to enhanced surveillance during pregnancy and delivery.

As expected, adenomyosis has shown correlation with endometriosis and infertility, as evidenced by the increased employment of assisted reproductive techniques [5,6,28]. Furthermore, a higher frequency of placenta previa has been reported in the adenomyosis group. By altering the uterine environment through anatomical, functional and immunochemical mechanisms, adenomyosis has been proposed as a contributing factor in the pathophysiological link between endometriosis and abnormal placentation [9,29,30,31,32,33,34]. Both endometriosis and adenomyosis may disfigure the endometrial cavity and this might concur to the inefficient function of the uterus at the time of delivery. Besides, adenomyosis shares major risk factors with placenta accreta spectrum disorders. Probably, no correlation emerged in this study because the population was highly selected and there was a high frequency of previous caesareans in both groups. In our opinion, further investigations in this area would be valuable [9].

The correlation between adenomyosis and previous hysteroscopic metroplasty is an intriguing finding. Since this type of surgery directly involves the endo-myometrial junctional zone, it could be hypothesized as having a role in adenomyosis etiology as for other interventions such as caesarean section or curettage for abortion [15,35,36]. In addition, several cases of uterine rupture have been reported after complicated and uncomplicated hysteroscopic metroplasty, suggesting a possible role in the uterine wall weakening [37,38]. In our series, we reported one case of uterine rupture following hysteroscopic metroplasty in the adenomyosis group, and one in the unaffected group. However, comparative retrospective studies have failed to confirm a clear correlation between this surgical procedure and obstetric complications in subsequent pregnancies [39,40,41]. In our opinion, even if a causal contribution of this surgery in the genesis of adenomyosis is plausible, the presence of both factors could synergistically increase the risk of obstetric complications regardless of their reciprocal correlation. Further evidence on this issue would be valuable.

Furthermore, the correlation between adenomyosis and previous pelvic surgery in general deserves to be discussed. Given the study design, it is not possible to disentangle whether the increased frequency of prior surgery for endometriosis and metroplasty played a role in the genesis of adenomyosis, or directly cogenerated the complications that led to PH. However, surgery for endometriosis usually doesn’t involve the myometrial layer. Besides, we did not observe a significant difference between the two groups in terms of previous surgery with higher myometrial impact, such as previous caesarean and myomectomy. Therefore, while assuming that in some cases adenomyosis may recognize a postsurgical origin, we can reasonably presume that in our study the differences observed were primarily due to the presence of the disease rather than to previous surgery.

Previous studies reported higher risk of preterm birth and preterm premature rupture of membranes (pPROM) associated with adenomyosis [5,6,7,8]. Consistent with these findings, cases of adenomyosis in our study showed a one-month lower gestational age at birth compared to the unaffected group. Even if the association with pPROM was not confirmed, an indirect correlation may be derived from the more frequent histological evidence of chorionamnionitis among the affected cases. Multiple factors could have contributed to this finding, including a combination of pPROM, antepartum bleeding, and intrauterine fetal death.

Restricting the analysis to twin gestations, we registered a higher frequency of intrauterine fetal demise among patients with adenomyosis, confirming a recent report by Kim et al. [42].

Some main strengths and limitations of this study deserve to be considered. The high-volume institution and the 12-years study period allowed a comprehensive analysis and inclusion of a relevant number of PH, which is a relatively infrequent situation. As a referral hospital, the availability of gynecological pathologist ensured adequate experience in identifying the disease.

The standard analysis of surgical specimens may have played an ambivalent role. It certainly has the advantage of pursuing a reproducible method, allowing investigation over a long period. On the other hand, standard sampling technique could potentially have missed some cases of focal adenomyosis, in particular adenomyosis of the outer myometrium. As a consequence, we were not able to provide a reliable distinction between different phenotypes of adenomyosis. In agreement with the literature derived from the analysis of gynecological specimens, we believe that a prospective investigation of obstetric cases based on the inclusion of multiple samples could lead to the detection of a higher prevalence of adenomyosis [15,16,21]. In our opinion, the most important limitation of the study is the generalizability of results. The study population consisted of women who had undergone hysterectomy, which is a very rare event in obstetric care. It is usually performed as an emergency life-saving procedure when conservative treatments have failed. Furthermore, the experience of a single referral hospital is undoubtedly influenced by the higher concentration of high-risk obstetric cases when compared to the general population. These issues may limit the generalizability of our findings. Finally, the retrospective design is unable to clarify cause-effect relationships.

## 5. Conclusions

Adenomyosis is associated with more challenging obstetric and perinatal outcomes among women undergoing PH. Even if a causal relation cannot be firmly established (previous surgery may be a confounder), it can nonetheless be inferred that failed diagnosis of adenomyosis may hamper clinical awareness of enhanced obstetric risk. Adequate pregestational and prenatal counselling for women suffering from this condition is mandatory. Future investigations should focus on the prospective assessment of the impact of adenomyosis on obstetric care.

## Figures and Tables

**Figure 1 jcm-11-04814-f001:**
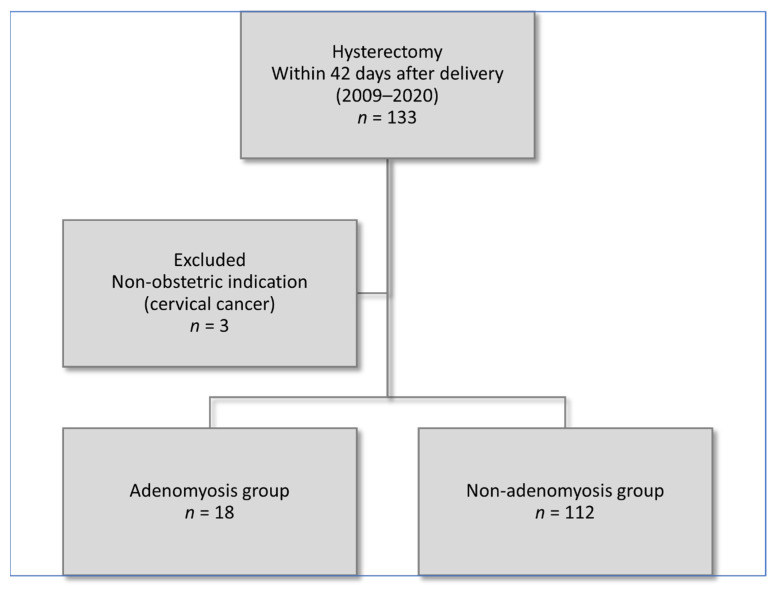
Flow diagram of study population.

**Figure 2 jcm-11-04814-f002:**
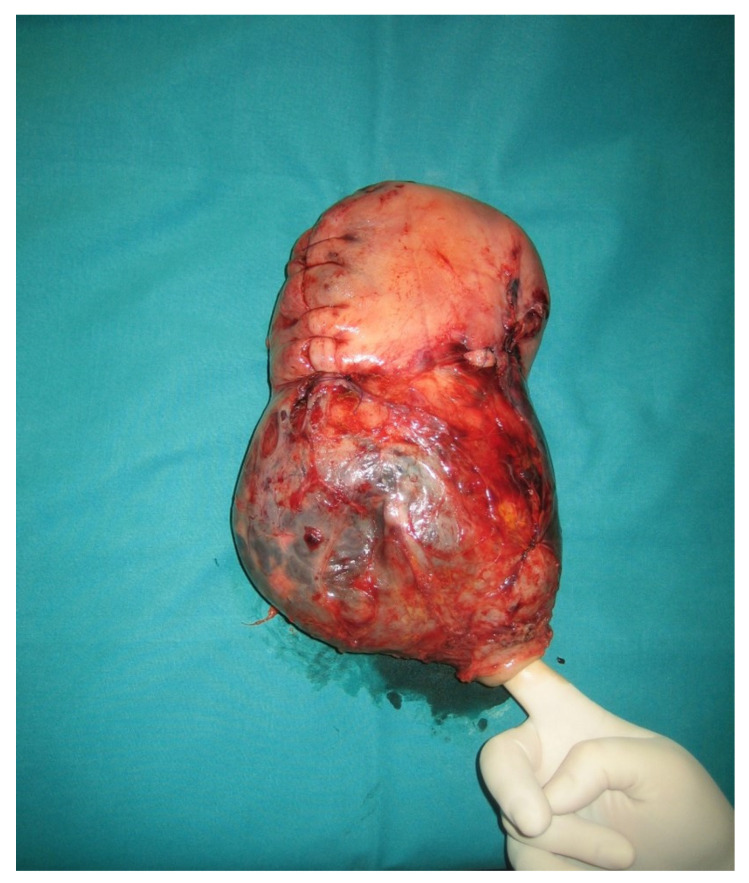
Macroscopic surgical specimen. The uterine cervix is indicated by the surgeon’s finger on the lower side. Above the cervix, the lower uterine segment is bulging because of the presence of placenta previa accreta with evidence of reduced myometrial thickness and massive neovascularization. On the upper uterine corpus, longitudinal caesarean section scar avoiding the placental edge is shown.

**Figure 3 jcm-11-04814-f003:**
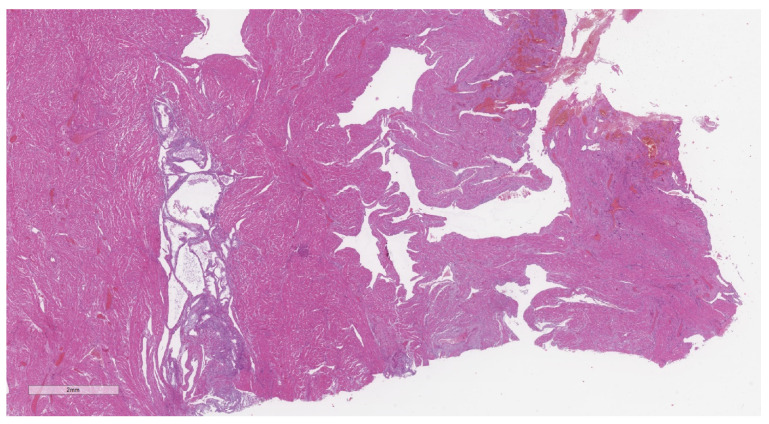
Representative microphotograph of a post-partum uterus with adenomyosis. Hematoxilin and eosin (H&E), 1×. At low-power magnification, the placental implant is evident on the right, and with the underlying adenomyosis on the left.

**Figure 4 jcm-11-04814-f004:**
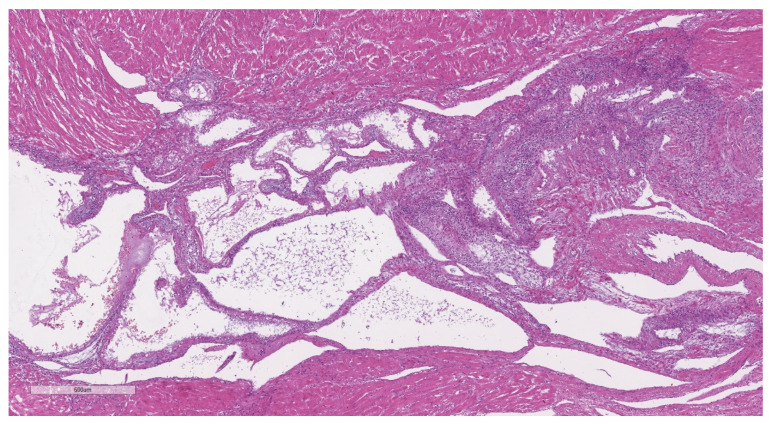
H&E, 4×. The decidualized adenomyosis focus demonstrated large endometrial glands, some with ectasia, and an underlying endometrial stroma.

**Figure 5 jcm-11-04814-f005:**
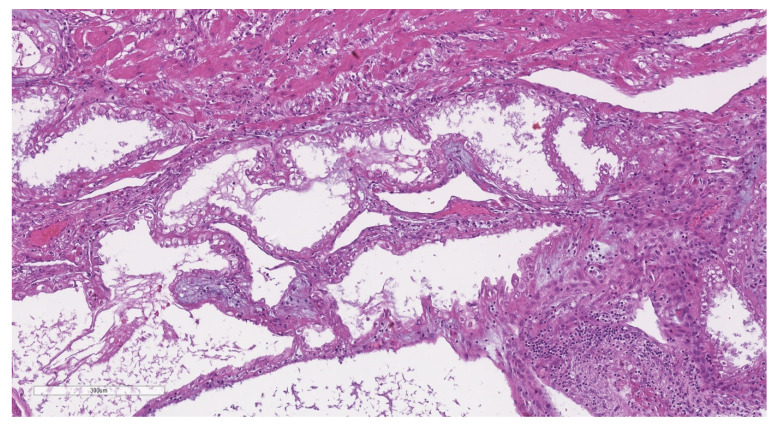
H&E, 10×. The glandular cells of the decidualized endometrium in the adenomyosis foci shows hypersecretive features, such as a clear cytoplasm.

**Figure 6 jcm-11-04814-f006:**
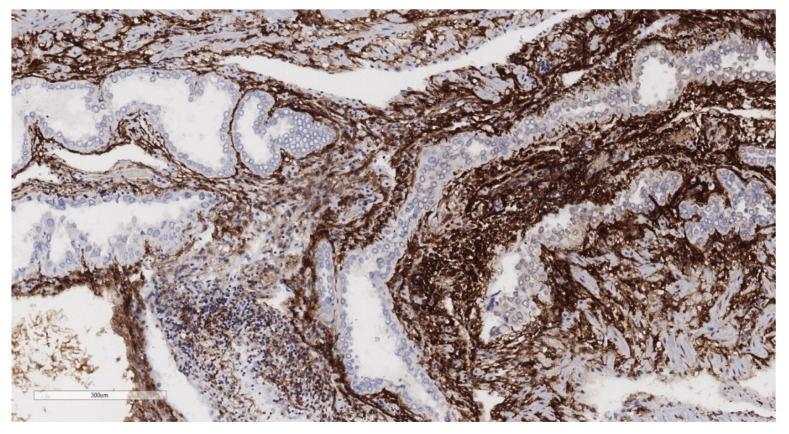
CD10, 10×. Immunohistochemistry for CD10 highlights the endometrial stroma.

**Table 1 jcm-11-04814-t001:** Baseline characteristics of the study population stratified by presence or absence of adenomyosis.

Characteristics	Adenomyosis		Non-A.		*p*-Value
	*n* = 18	% or SD	*n* = 112	% or SD	
Maternal age (years)	39.7	5.2	37.6	4.5	0.08
Country of origin					
Italy	13	72.2	71	63.4	0.47
Others	5	27.8	41	36.6	
Level of education					
Lower level	5	27.8	12	10.7	0.1
High school	4	22.2	25	22.3	0.76
University education	6	33.3	45	40.2	0.58
Missed	3	16.7	30	26.8	
Occupational status					
Working	13	72.2	78	69.6	0.78
No-working	3	16.7	15	13.4	
Missed	2	11.1	19	17.0	
Marital status					
Married	12	66.7	78	69.6	0.8
Unmarried	6	33.3	34	30.4	
Obstetric history					
Nulliparous	15	83.3	84	75.0	0.44
Previous vaginal delivery	3	16.7	28	25.0	0.45
One previous caesarean	6	33.3	25	22.3	0.31
Two or more previous caesarean	4	22.2	27	24.1	0.86
Previous curettage for abortion	8	44.4	32	28.6	0.17
Gynecological disease					
Previous surgery for uterine fibroid *	2	11.1	6	5.4	0.35
Uterine fibroid **	4	22.2	7	6.3	0.07
Previous surgery for endometriosis ***	3	16.7	3	2.7	0.004
Previous adenomyomectomy ****	1	5.6	0	0.0	0.13
Endometriosis **	5	27.8	9	8.0	0.035
Previous hysteroscopic metroplasty for uterine septum *****	3	16.7	2	1.8	0.017

Data are expressed as mean ± SD or Number and %. * Surgical technique (Adenomyosis/Non-A. group): Laparotomy (2/4); Laparoscopy (0/2); the endometrial cavity was involved in two cases of the Non-A group. ** Inclusion by medical history and post-PH histology. *** Surgical technique (Adenomyosis/Non-A. group): Laparotomy (1/0); Laparoscopy (2/3). **** Surgical technique: Laparotomy; the endometrial cavity was not involved. ***** Surgery was reported as unremarkable (no complications) in all cases.

**Table 2 jcm-11-04814-t002:** Pregnancy outcome stratified by presence or absence of adenomyosis.

Characteristics	Adenomyosis		Non-A.		*p*-Value
	*n* = 18	% or SD	*n* = 112	% or SD	
Pregnancy complications					
Twin gestation	4	22.2	23	20.5	0.87
Assisted reproductive technology	9	50.0	22	19.6	0.01
Hypertensive disorder of pregnancy	1	5.6	10	8.9	0.71
Placenta previa	14	77.8	51	45.5	0.01
Placenta accreta spectrum	12	66.7	60	53.6	0.3
Preterm premature rupture of membrane	2	11.1	6	5.4	0.34
Antepartum bleeding	3	16.7	10	8.9	0.31
Mode of delivery					
Vaginal delivery	0	0.0	15	13.4	0.13
Elective caesarean	12	66.7	63	56.3	0.41
Emergency caesarean	6	33.3	34	30.4	0.8
Clinical indication for hysterectomy					
Haemorrhage: placenta previa or accreta spectrum	11	61.1	57	50.9	0.3
Haemorrhage: uterine atony	4	22.2	49	43.8	0.14
Uterine rupture	2	11.1	3	2.7	0.28
Sepsis	1	5.6	3	2.7	0.51
Maternal and fetal outcomes					
Estimated blood loss (ml, SD)	4072	2055.0	4726	3195.0	0.41
Packed red cells units transfused	7.3	5.6	9.5	6.4	0.17
Major surgical complications *	5	27.8	20	17.9	0.34
Medical complications **	0	0.0	8	7.1	0.59
Gestational age at delivery (weeks, SD)	32	4.6	35.5	3.6	0.0004
Small for gestational age at birth	2	11.1	7	6.3	0.6
Chorionamnionitis	5	27.8	6	5.4	0.008
Stillbirth	2	11.1	1	0.9	0.07
Intrauterine fetal demise in twin pregnancy	2	50.0	1	4.5	0.048

Data are expressed as mean ± SD or Number and %. * Urinary tract lesion, relaparotomy. ** Disseminated intravascular coagulation, haemorrhagic shock, kidney failure.

## Data Availability

M.O. had access to the complete dataset used in the study and takes responsibility for the integrity of the data and accuracy of the data analyses. The dataset is available upon justified request.

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
