# Peer review of "The Prevalence and Clinical Impact of Adenomyosis in Pregnancy-Related Hysterectomy"

_jcm, 2022, doi:10.3390/jcm11164814_

Round 1

Reviewer 1 Report

Thank you for this nice paper, However need further clarifications on certain aspects to make the data more clear. It would make the paper even better.

Line 1: Please identify the article variety whether is a data analysis

Line 32: Please consider to add few words from comments section on Lines 149-151

Line 69: however this is given in the figure 1 is it necessary?

Table 1 and  Lines 101 -102, 122:  Although these cases were highlighted amongst the table . it shows 18 cases of previous uterine surgeries. of which  8 are for endometriosis related. 

1. It is not clear as if there are any overlaps between the cases; that is if any of the patients had more than 1 surgeries that was  enlisted in Table1?

2.  All the mentioned surgeries are highly prone for Placenta accreta - which accounts for 12 cases

3. It is also need clarification if the Uterine septal defect correction involved over correction into the Fundal Myometrium, which is obvious for the 1 case of Uterine rupture.

4. Whether the mentioned cases of adenomyosis with uterine septum had concomittant adenomyosis or endometriosis surgery , 

5. Whether or not the classification of septal defect was exactly met for criterion of treatment.

6. It is also not uncommon for endometrial cavitory disfuguration with severe endometriosis or adenomyosis.

7. Curettage is also a significant reason for Placenta accretal disorders.

8. It was  not mentioned the routes of surgery for Fibroid, Endometriosis or adenomyosis, which could also be a detrimental factor for the outcome.

9. There is 1 case of adenomyomectomy and 4 Myomectomy , do we have a clarity if the endometrial cavity was breached and on which closure technique was employed 

10. if the myometrial scars and myometrial healing were observed post operatively in these cases?

Lines 149-151: Is there any data on the fact that in the series, there is no clear evidence to differentiate if these patients had adenomyosis as a result of previous surgeries or if the disease itself resulted n the outcome. 

If not, it might be good to discuss about that here as well.

Conclusion Lines 203-204: It would also be good to mention about the risk of the surgeries for adenomyosis, septum and Endometriosis  as mentioned in the result discussion table 1, given the matter of fact that there is no clear evidence if all the PH resulted exclusively from adenomyosis. 

Reviewer 2 Report

Dear authors, 

thank you for submitting this very interesting retrospective cohort study. 

I just have a few comments on your manuscript: 

Introduction:

- Adenomyosis is frequently asymptomatic? I do not agree. Most of patients with adenomyosis are symptomatic and it seems that many of very young symptomatic patients have Adenomyosis that we still can't diagnose properly. What does frequently mean? Please explain.

- I agree that TVS is the diagnostic method of first choice. Maybe you could add this citation that supports your statement. 

Tellum T, Nygaard S, Lieng M. Noninvasive Diagnosis of Adenomyosis: A Structured Review and Meta-analysis of Diagnostic Accuracy in Imaging. J Minim Invasive Gynecol. 2020 Feb;27(2):408-418.e3. doi: 10.1016/j.jmig.2019.11.001. Epub 2019 Nov 8. PMID: 31712162.

Line 48: Here it could be suitable to mention the problems of obtaining a reliable biopsy.

Histopathology: As I understand, you concentrated on the endometrial-myometrial layer. What about the outer myometrium then?

In think the lack of classification of the disease and the missing understanding of the different phenotypes could be discussed.
